# Performance Comparison of Quantized Control Synthesis Methods of Antenna Arrays

**David Pánek** [1,*] , **Tamás Orosz** [2] , **Pavel Karban** [1] , **Deubauh Cedrick D. Gnawa** [1]
**and Hamid Keshmiri Neghab** [1]

1   Department of Theory of Electrical Engineering, University of West Bohemia, 306 14 Plzeň, Czech Republic;
    karban@kte.zcu.cz (P.K.); gnawa@fel.zcu.cz (D.C.D.G.); neghab@fel.zcu.cz (H.K.N.)
2   Faculty of Engineering Sciences, Department of Automation, Széchenyi István University of Győr,
    Egyetem tér 1, H-9026 Győr, Hungary; orosz.tamas@sze.hu
*   Correspondence: panek50@fel.zcu.cz; Tel.: +420-377-634-657

**Abstract:** There is a great potential in small satellite technology for testing new sensors, processes, and technologies for space applications. Antennas need careful design when developing a small satellite to establish stable communication between the ground station and the satellite. This work is motivated by the design of an antenna array for a future rotatorless base station for the VZLUSAT group of Czech nano-satellites. The realized antenna array must cover a relatively broad range of elevation and azimuth angles, and the control must be fast enough to track the satellite in low Earth orbits. The paper deals with possibilities of synthesis of quantized control of the antenna array. It compares quantization influence for well-known deterministic synthesis methods. It shows the method for decreasing computational cost of synthesis using optimization approach and presents the multi-criteria optimization as a tool for reaching required radiation pattern shape and low sensitivity to quantization at the same time.

**Keywords:** antenna array; synthesis control; quantized control; array factor

## 1. Introduction

The CubeSat Launch Initiative provided an attractive opportunity for universities, high schools, and non-profit organizations to build small satellites, which can fit an N-Unit cubic structure [1–3]. Here, each unit of the CubeSat should fit an $N \times 1$ dm$^3$ cube (=1U size), which can contain one or more systems of the satellite. The common CubeSats has a very small 1U, 2U, or 3U size [4–7]; due to the recent upgrade of the standard, the largest satellites can reach a 27U size. These satellites are launched to Low-Earth-Orbit (LEO) with 250–900 km altitude above the ground. Most of the CubeSats use Radioamateur frequency band for the main communication [8], but licensed bands are now often used due to the commercial nature of some missions. Due to the reduced launch costs and the inspiring increase in complex applications, this platform started to attract many commercial, military, and governmental organizations [6,9]. These applications increased the need for larger transfer rates. The Ka-band started to be explored as a possible solution besides the S and X bands.

An interesting solution to this problem can be achieved by antenna array technology, which can have many advantages over parabolic antennas. From a mechanical point of view, it does not require design and maintains a drive system, which sets the azimuth and the elevation angles. Such systems have a simpler feeding network that cannot be disconnected during the connection time. These tools are insensitive to the moisture and weather conditions during the mission. Moreover, with a pattern reconfigurability algorithm, they can support multi-task missions [6,10].

Although the field of antenna array design and control synthesis have broadly been studied for many decades, there are still many sub-domains that have received less attention

so far. One of such sub-domains is their robust design, which takes into account various uncertainties in their design and suppresses their influence. The robust design is becoming more and more important together with the development of new technologies which allow creating antennas on flexible and stretchable materials such as Aerosol Jet Print [11–13]. In specific applications, such as printing antennas on flexible materials which can be bent and placed into the capsule of in-body antenna, printing patterns on curved surfaces of in-body antennas, or incorporating antennas into wearable electronics, it is important to study the influence of the change in geometry on the system properties [14–16].

The second area that has received little attention is the design of quantized control of the antenna array [17]. The recent review dedicated to the control synthesis methods was published in [18]. This paper offers a detailed analysis of methods, especially in connection with the 5G communication systems. The advantage of quantized control of antenna array consists mainly in the potential for fast and precise reconfiguration of the antenna array beam direction, which is important for tracking satellite position. Within the field of antenna array synthesis, a number of numerical techniques are commonly used, which can be divided into three groups:

- deterministic methods,
- optimization techniques,
- machine-learning tools.

Deterministic methods mean the two main classes of the array control synthesis methods [19–21]. The first class uses electromagnetic field simulation. The simulation takes into account a complex model of the antenna array including mutual coupling and also feeding circuits. The second class starts from the analysis of the array factor that is based on the assumption that all elements of the antenna array are identical and mutual couplings between particular elements are negligible.

Many papers aimed at the optimization methods for the control synthesis [22–26] have already been published. It is possible to say that most of the published approaches are based on evolutionary methods such as Genetic Algorithms (GA), Particle Swarm Optimization (PSO), Simulated Annealing (SA), and Ant Colony Optimization (ACO) [27]. Due to the high computational cost of the proposed methodologies, great efforts were also devoted to reducing the computational complexity. For example, paper [28] takes advantage of the Taguchi's method for control linear array control synthesis. Ref. [19] proposed the use of SMPSO for sparse antenna array synthesis. In the paper, the performance is compared with differential evolution and genetic algorithm. The main drawback of these techniques is that they usually consider only the array factor with the aid of deterministic methods, but neglect the mutual coupling between particular between array elements. The machine learning-based tools play an increasingly important role to resolve this problem and design smart antennas (antenna arrays with dynamic control) [29].

Like most of the CubeSat projects [8], VZLUSAT uses a radio-amateur frequency band 435–438 MHz (70 cm) for communication [6] and the satellites are deployed on a Low Earth Orbit (LEO), commonly ranging from 350 km to 900 km from the Earth surface. This particular application require specific demands to be satisfied to ensure the maximal contact time with the satellite. For one ground station, two or up to four long contacts with good communication conditions are possible in a day. These long contacts usually take no more than 15 min. From these data, the maximum communication distance between the ground station and the satellite can be calculated. It varies from 2000 to 3500 km, while the orbit revolution time is 91 to 103 min. The transmission to the satellite generally starts above 5° elevation to avoid interference with terrestrial systems. The Doppler shift and relative position of LEO satellites to the ground station are changing rapidly during contacts in high elevation angles, which increases the demand on the antenna positioning system [4,6].

The current VZLUSAT-1 and VZLUSAT-2 nanosatellite projects inspired the proposed application [4,5] to design an antenna array for communication with the future small satellites. The main goal of the work is to find a methodology for the synthesis of control for the antenna arrays, which is robust in the sense that the effect of perturbations in the excitation

is minimized. The proposed methodology will be compared with a small theoretical example. The goal of the comparison is to select the most appropriate methodology, which can be used together with the more complex, 3D FEM-based calculations for the more detailed antenna analysis. The proposed methodologies and the source code of the model can be accessed from the homepage of the project (https://github.com/panek50/pyntenna, accessed on 20 March 2022).

## 2. Control Synthesis

The problem of control synthesis is closely related to the design of the antenna array itself, and it was divided into three steps:

- Preliminary design—the control is designed according to application requirements, usually using well-known deterministic methods. In this step, implementation details, such as connecting circuits and electronic parts, are not considered in the field models. Therefore, the results obtained using simplified methods (such as the array factor approach described below) are relatively good with the field models.
- Design—the field models contain all important details. The results of full-field simulations differ from simplified methods. In this step, optimization methods and methods based on matrix inversions are usually used.
- Implementation and calibration the results obtained by measuring manufactured hardware differ from results obtained using full-field simulation.

### 2.1. Model Example

As a model example, the control synthesis for the application of the base station for nanosatellites was chosen. This application brings certain requirements for the antenna array:

- As a compromise between the complexity of control and the required gain of the antenna array, the size of the array was chosen to be $11 \times 11$.
- The antenna array is designed for the S band and particular frequency $f = 2.405\,\text{MHz}$. The required bandwidth is $50\,\text{GHz}$.
- The required elevation steering angle is $\Delta\theta = 30°$ and the required steering azimuth angle is $\Delta\phi = 30°$.
- The maximal radiated power is $0.5\,\text{W}$ per patch; the overall radiated power is $P = 60.5\,\text{W}$, which corresponds to the power radiated from currently used parabolic antenna with the diameter $2\,\text{m}$.
- The minimal beam-width is $6°$.

The synthesis will be based on the candidate design of the antenna element. The element was designed using Antenna Magus 2020 software [30]. The shape of the element is shown in Figure 1 and the specific dimensions are summarized in Table 1.

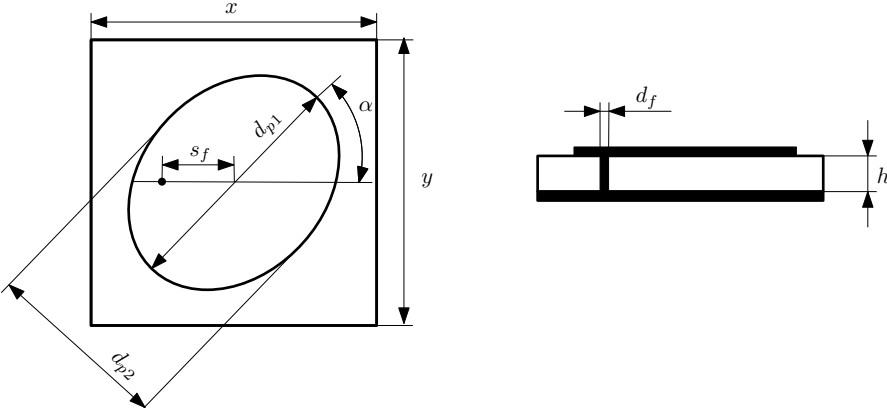

**Figure 1.** Shape of the antenna array element.

**Table 1.** Dimensions of designed patch (distances are in millimeters).

| $f$ [MHz] | $Z_{\text{in}}$ [Ω] | $x$ | $y$ | $\alpha$ | $d_{\text{f}}$ | $d_{\text{p1}}$ | $d_{\text{p2}}$ | $s_{\text{f}}$ |
|---|---|---|---|---|---|---|---|---|
| 2.405 | 50 | 48.2 | 48.2 | 45° | 0.91 | 49.2 | 47.16 | 8.4 |

The directivity of one patch defined as

$$D(\theta, \varphi) = 4\pi \frac{P_{\text{U}}}{P_{\text{T}}},\tag{1}$$

where $P_{\text{U}}$ stands for the power radiated per unit of the solid angle and $P_{\text{T}}$ means the total radiated power, is depicted in Figure 2. As mentioned above, the considered array consists of 11 × 11 elements. The distance of centers of two elements in $x$-direction and $y$-direction is

$$d_x = d_y = \frac{\lambda}{2} = \frac{c}{2f} = \frac{3 \cdot 10^8}{2 \cdot 2.405 \cdot 10^9} \approx 0.062\,\text{mm}.\tag{2}$$

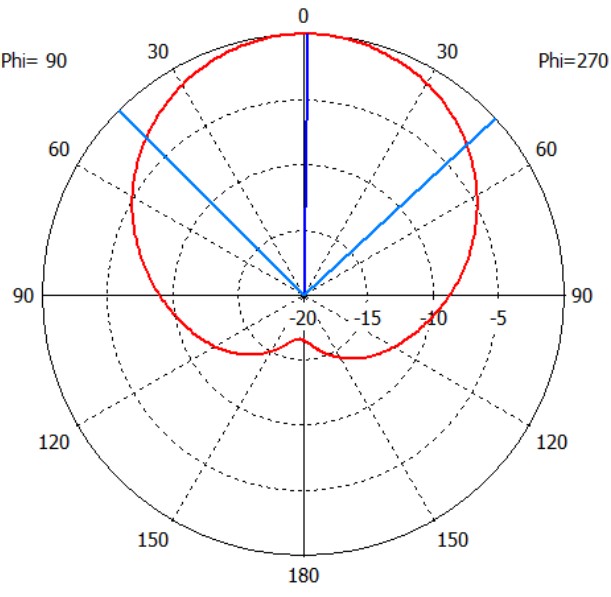

Farfield (Array) Directivity Abs (Phi=90)

Theta / Degree vs. dB

**Figure 2.** The diagram shows the directivity of one antenna array element.

The model of the antenna array is depicted in Figure 3a, and the corresponding directivity pattern for uniform excitation is depicted in Figure 3b.

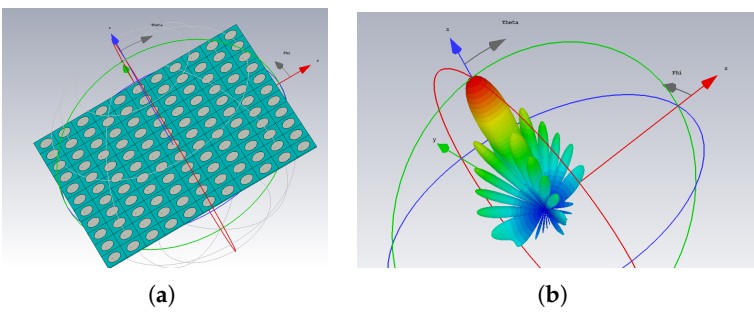

(a)　　　　　　　　　(b)

**Figure 3.** Figure (**a**) shows the model of the antenna array in CST Studio, while picture (**b**) plots the directivity pattern of the proposed antenna array.

### 2.2. Array Factor

As the full field solution for larger antenna arrays is often too computationally complex, it is often used simplified using the array factor. The electromagnetic field produced by the antenna array can be described using electric field strength $E$ and magnetic field strength $H$, which for a harmonic source, satisfies the equations

$$\Delta \underline{H} + \underline{k}^2 \underline{H} = \underline{0}, \qquad \Delta \underline{E} + \underline{k}^2 \underline{E} = \underline{0}, \tag{3}$$

where $\underline{k} = -j \cdot \omega \mu (\gamma + j \cdot \omega \varepsilon)$.

The vector of electric field strength produced by antenna array $\underline{E}(\vartheta, \varphi)$ can be, at a particular point, expressed as [31]

$$\underline{E}(\vartheta, \varphi) = \underline{f}_{\mathrm{e}}(\vartheta, \varphi) \cdot S_{\mathrm{a}}(\vartheta, \varphi), \tag{4}$$

where $\vartheta$ stands for the elevation, $\varphi$ for azimuth, $\underline{f}_{\mathrm{e}}$ is the electric field produced by one element, and $S_{\mathrm{a}}$ is the array factor in the form

$$S_{\mathrm{a}}(\varphi, \vartheta) = \sum_{n=1}^{N} \sum_{m=1}^{M} \underline{I}_{nm} \exp\left[j \cdot k_0 \cdot n \cdot d_x \sin(\theta) \cos(\varphi) + k_0 \cdot m \cdot d_y \sin(\vartheta) \sin(\varphi)\right], \tag{5}$$

where $N$ is the number of elements in the $x$ direction, $M$ is the number of elements in the $y$ direction, $d_x$ and $d_y$ are the distances between the patches, and $k_0 = 2\pi/\lambda$ is the wave number. The dimensions are depicted in Figure 4. For other derived quantities, as directivity defined in Equation (1), the decomposition into the patch pattern and array factor works in the same way.

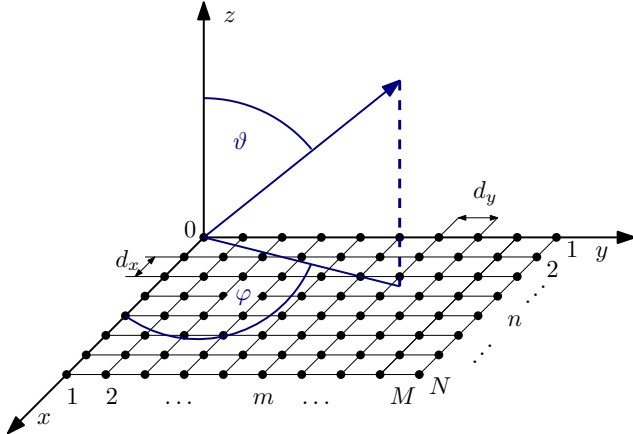

**Figure 4.** Planar antenna array with $N \times M$ elements.

The control synthesis based on array factor obviously cannot bring the optimal results, but it represents a suitable first step to obtain the point from which the optimization can start. It also can bring an idea about the physical limits which can be (at least theoretically) reached.

### 2.3. Matrix Formulation of Array factor

The computational complexity of the array factor calculation can be reduced using a matrix formulation in the form

$$S_{\mathrm{a}} = \boldsymbol{LI} = \begin{bmatrix} \alpha_{11} & \alpha_{12} & \cdots & \alpha_{1k} \\ \alpha_{21} & \alpha_{22} & \cdots & \alpha_{2k} \\ \vdots & \vdots & \ddots & \vdots \\ \alpha_{l1} & \alpha_{l2} & \cdots & \alpha_{lk} \end{bmatrix} \cdot \begin{bmatrix} I_{11} \\ I_{12} \\ \vdots \\ I_{lk} \end{bmatrix}, \tag{6}$$

where

$$\alpha_{mn} = \exp\left[ j \cdot k_0 \, m \, d_x \sin(\theta) \cos(\phi) + k_0 \, n \, d_y \sin(\theta) \sin(\phi) \right] \tag{7}$$

and $I$ is the vector of excitation of particular patches.

This formulation is advantageous for repeated calculations, while the computational complexity is removed in the assembling process. The complexity can consequently be reduced using the Principal Component Analysis (PCA) [32]. The correlation matrix $C$ of the transformation matrix $L$ can be calculated as

$$C = L \cdot L^{\mathrm{T}}. \tag{8}$$

Only a limited number of eigenvalues play the important role, as can be seen in Figure 5a. Figure 5b shows the comparison of array factor calculated from excitation obtained by direct usage of Dolph–Chebychev method and reconstructed by pseudo-inverse with PCA reduction.

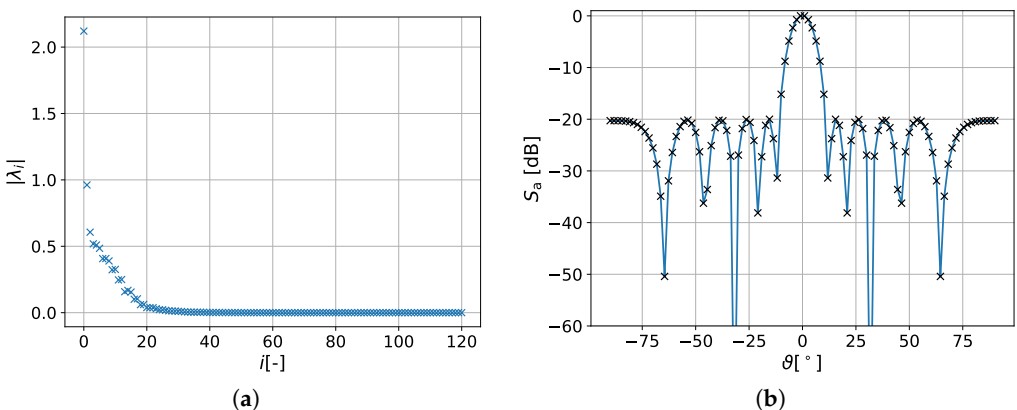

(a)  (b)

**Figure 5.** Order reduction and pseudo-inverse. (**a**) Eigenvalues of transformation matrix; (**b**) Comparison of array factor calculated from excitation obtained by direct usage of Dolph–Chebychev method and reconstructed by pseudo-inverse.

The reduced matrix can be also used within the Penrose–Moore pseudo-inverse for the control synthesis. The transformation matrix can be decomposed using Singular Value Decomposition (SVD) as

$$L = U \cdot \Sigma \cdot V^{\mathrm{T}}, \tag{9}$$

where $\Sigma$ is a diagonal matrix with singular values in the main diagonal and $U$, $V$ are orthogonal matrices containing singular vectors. The pseudo-inverse of this matrix $\Sigma^*$ contains inverse singular values on the diagonal. The pseudo-inverse of the transformation matrix can be then written as

$$L^* = V\Sigma^* U^{\mathrm{T}}. \tag{10}$$

The usage of matrix pseudo-inverse can be useful for large arrays, however it tends to become weakly-conditioned for low resolution in $\vartheta$ and $\varphi$.

### 2.4. Deterministic Control Synthesis Methods

The deterministic methods use the tools common in the area of *digital signal processing*. These methods can be divided into three main sub-classes [28]:

- synthesis based on possessing nulls of the characteristic polynomial, commonly used for suppressing particular noise from the given direction, represented by Schelkunoff zero-placement method;
- methods based on so called beam shaping, i.e. specifying beam sector pattern and inverse transformation to obtain excitation, represented by Woodward–Lawson method,
- methods for synthesis narrow beams and low side lobes, represented by the Dolph–Chebychev method.

### 2.4.1. Woodward–Lawson Method

This design method is based on inverse Discrete Fourier Transform. The excitation of particular elements of the antenna array given by [33] was modified (for reaching a better efficiency) to the form

$$I(n, m) = \sum_{k=1}^{K} \sum_{l=1}^{L} S_A \cdot \exp(jk_0 \cdot (\mathbf{N} \cdot \Delta_x[k, l] + \mathbf{M} \cdot \Delta_y[k, l])), \tag{11}$$

where $\mathbf{I}$ is a matrix with complex numbers representing excitation, $k_0$ is the wave number, $\mathbf{N}$ and $\mathbf{M}$ are pre-assembled matrices containing numbers $1 \ldots N$ and $1 \ldots M$, respectively, in rows, and symbols $\Delta_x$, and $\Delta_y$ stand for preassembled matrices where

$$\Delta_x = \begin{bmatrix} d_x \sin(\vartheta_0) \cos(\varphi_0) & d_x \sin(\vartheta_0) \cos(\varphi_1) & \ldots & d_x \sin(\vartheta_0) \cos(\varphi_K) \\ d_x \sin(\vartheta_1) \cos(\varphi_0) & d_x \sin(\vartheta_1) \cos(\varphi_1) & \ldots & d_x \sin(\vartheta_1) \cos(\varphi_L) \\ \vdots & \vdots & \ddots & \vdots \\ d_x \sin(\vartheta_K) \cos(\varphi_0) & d_x \sin(\vartheta_K) \cos(\varphi_1) & \ldots & d_x \sin(\vartheta_K) \cos(\varphi_L) \end{bmatrix}. \tag{12}$$

and matrix $\Delta_y$ is built in a similar way.

### 2.4.2. Schelkunoff's Zero-Placement Method

The array factor (expressed in the $Z$-transformation) produced by $N \times M$ elements antenna array can be expressed as a product of $N - 1$ and $M - 1$ degree polynomials using its roots (zeros) in the form (extended from the 1D version published in [33])

$$S_a(z) = \sum_{n=0}^{N-1} \sum_{m=0}^{M-1} a_n b_m z^m z^n = \tag{13}$$

$$(z - z_1)(z - z_2) \ldots (z - z_{N-1})a_{N-1} \cdot (y - z_1)(y - y_2) \ldots (y - y_{M-1})b_{M-1}.$$

The magnitudes and phases on the antenna array can be expressed as a dyadic product of two complex vectors

$$\mathbf{I} = \mathbf{a} \otimes \mathbf{b}, \tag{14}$$

where $\mathbf{a} = (a_0, a_1, \ldots, a_{N-1})$ and $\mathbf{b} = (b_0, a_1, \ldots, b_{M-1})$ are coefficients of polynomials from equation (13). The first step of the method is proper placing of zeros $z_n$ and $y_n$ in the $\vartheta$ direction or $\varphi$ direction, respectively. Consequently, the coefficient of polynomials $a_n$ and $b_m$ are calculated. The excitation matrix is then obtained directly using the dyadic product from equation (14).

### 2.4.3. Dolph–Chebychev Method

It can be proven that the lowest main-lobe width and highest ratio of suppression of side lobes at the same time can be achieved if all side lobes have the same level. This can be achieved using the Dolph–Chebychev window which is based on the Chebychev polynomials. The Chebychev polynomial of degree $n$ is given by expression

$$P_n(x) = \cos(n \arccos(x)). \tag{15}$$

The procedure itself is similar to the zero-placement method, but the position of zeros in the $z$-plane is given by zeros of the Chebychev polynomials as

$$z_i = \exp(j \cdot \psi), \qquad y_i = \exp(j \cdot \zeta), \tag{16}$$

where

$$\psi = 2 \cdot \arccos\left(\frac{\alpha_n}{\alpha_0}\right), \tag{17}$$

$$\zeta = 2 \cdot \arccos\left(\frac{\beta}{\beta_0}\right). \tag{18}$$

There holds

$$\alpha_0 = \cosh\left(\frac{\operatorname{arccosh}(R_a)}{N-1}\right), \qquad \beta_0 = \cosh\left(\frac{\operatorname{arccosh}(R_a)}{M-1}\right), \tag{19}$$

where $R_a$ is the required suppression of side lobes level,

$$\alpha_n = \cos\left(\pi \cdot \frac{n-0.5}{N-1}\right), \qquad \beta_n = \cos\left(\pi \cdot \frac{m-0.5}{M-1}\right). \tag{20}$$

### 2.5. Optimization Methods

Computational complexity of control synthesis can be divided into two main groups: computational complexity during the synthesis itself and computational complexity during the real deployment. Regardless of the numerical techniques used, the measure of complexity is the required number of array factor calculations or electromagnetic field simulations. It seems that the pure usage of optimization tools for array factor synthesis does not bring any significant advantage. Although the optimization tools seem to provide more freedom in choosing the target, there is a certain equivalence between the optimization with a particular goal function and some deterministic approach. For example, the optimization where the goal is to maximize the magnitude of main lobe gives the same result as the Woodward–Lawson method with a rectangular window (see Figure 6).

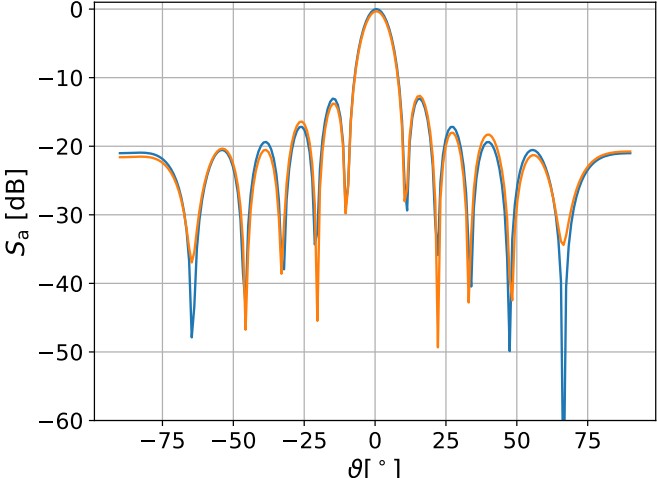

**Figure 6.** Comparison of Woodward–Lawson method (orange line) and optimization (blue line).

The most computationally expensive part of the control synthesis is an optimization procedure, provided it is used. Although the results from the array factor analysis are presented in the paper, the goal is to find an approach that will work together with electromagnetic field simulation. This fact limits the number of calculating goal functions to hundreds or lower thousands. The paper deals with testing three different sets of optimization parameters:

- Magnitude and phase of each particular element represent the optimization parameters.
- Coefficients of polynomials $a, b$ from Equations (13) and (14) are parameters of optimization.
- Positions of zeros $y_i$ and $z_i$ from Equation (13) are subject of optimization.

In preliminary tests, it appeared that gradient-based methods and methods following from convex optimization such as Nelder–Mead, COBYLA, BOBYQA showed poor convergence. Surprisingly, bad convergence also appeared by methods commonly used in machine learning such as Covariance Matrix Adaptation Evolution Strategy (CMA-ES). Together with the requirement on the possibility of multi-criteria optimization as promising candidates appeared methods based on genetic algorithms NSGA-II, EPS-MOEA, and swarm optimisation algorithm SMPSO. The aim was to assess the speed of convergence, especially at the end of the optimization process. This is based on the idea of a hybrid algorithm, where the starting point (population) of optimization is constructed on the basis of deterministic methods or methods based on electromagnetic field simulation.

*2.6. Quantization*

There are several approaches to quantization in the microwave technology. The particular approach depends on practical realization. In this paper, the quantization is performed on the level of array factor, which corresponds with the aimed realization using digitally controlled Variable Gain Amplifiers (VGA). The quantized magnitude $\hat{I}_{nm}$ of the coefficient $\underline{I}_{nm}$ represents the control of the gain of amplifier, the quantized phase $\hat{\varphi}$ represents the digital control of phase shifters. The quantization is performed using formula

$$\hat{I}_m = \lfloor I_m \cdot 2^n \rfloor / 2^n,$$ (21)

where $\hat{I}_m$ is the quantized weight, $I_{mn}$ is the normalized weight and symbols $\lfloor \rfloor$ are used for rounding.

$$\hat{\varphi} = \lfloor \varphi / \pi \cdot 2^n \rfloor / 2^n \cdot \pi.$$ (22)

The quantized magnitude and the quantized phase are directly used in simulations as weights for control excitation, and in practical realization they are interpreted as *n*-bits words used for control of VGA or phase sifters, respectively.

## 3. Results and Discussion

The first part of the study is dedicated to the comparison of effect of quantization for selected well-known methods. The influence of quantization on the array factor obtained by deterministic methods is depicted in Figure 7.

It is evident that the sensitivity strongly depends on the selected way of synthesis (All results presented in this section were obtained using software packages Pyntenna [34] and Artap [35,36], which are available at GitHub https://github.com/artap-framework/artap, accessed on 20 March 2022). It seems that the most robust method from the deterministic ones is the Woodward–Lawson method with a rectangular window (Figure 7a). The suppression of side lobes can be reached using windowing (Figure 7b), but application of a window increases the sensitivity to quantization. In the case of the Dolph–Chebychev method (Figure 7c), the sensitivity to quantization depends on the chosen level of suppressing the side lobes, the effect of quantization is greater with the greater required suppression of side lobes. The design using Schelkunoff's zero-placement method (Figure 7d) is an example of the design which could be acceptable for a continuous control but which is extremely sensitive to quantization (The position of zeros was chosen on purpose to demonstrate effect of quantization; we do not state that this is property of Schelkunoff's zero-placement method).

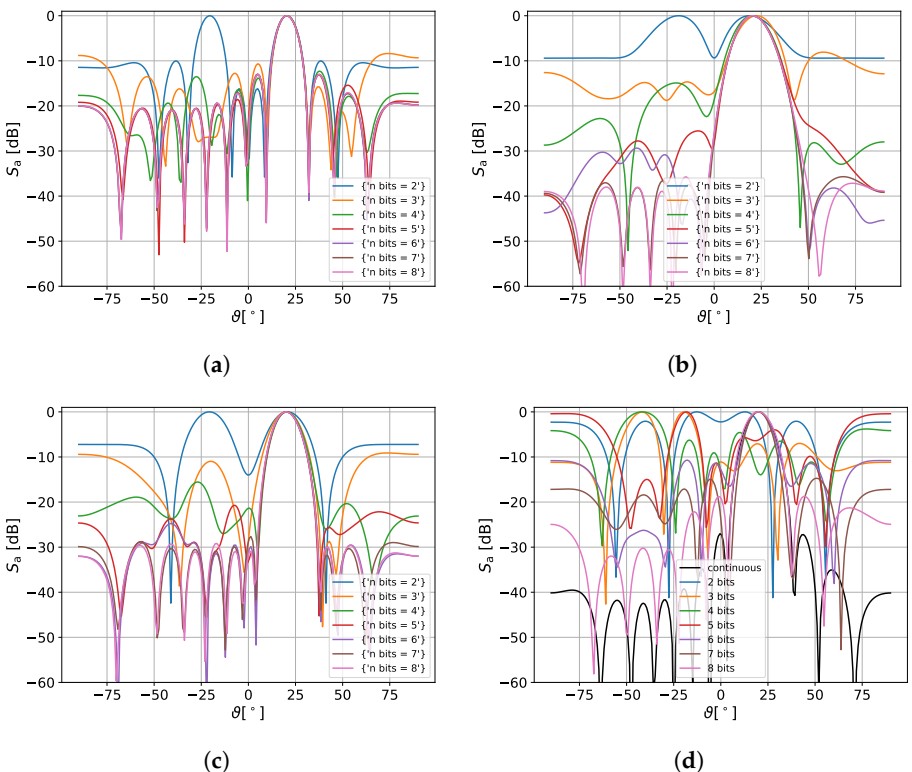

**Figure 7.** The influence of quantization of coefficients. (**a**) Fourier method with rectangular window; (**b**) Fourier method with Hamming window; (**c**) Dolph–Chebychev method with required side lobe level −30 dB; and (**d**) Schelkunoff's zero-placement method.

The dependence of the array factor on the scanning angle together with the effect of quantization is depicted in Figure 8. In all graphs in the figure, there is elevation angle $\vartheta$ on the *x*-axis, and on the *y*-axis is

$$\Delta = S_{a1} - S_{a2}, \tag{23}$$

where $S_{a1}$ stands for level of the main lobe and $S_{a2}$ represents the level of biggest side lobe. Figure 7 demonstrates that there is a connection between side lobe suppression and robustness against quantization error. Within the antenna array design process, it is necessary to carefully consider which criteria are more relevant for a given application. In the case of ground stations for nanosatellites, requirements on side lobes are relatively weak (compared to radar applications) but the robustness plays an important role. If steering angle is considered in a certain range, the effect of quantization is more noticeable than for a single elevation angle. The less sensitive design is reached using the Fourier method with a rectangular window (Figure 8a), where quantization using with more than four bits is acceptable. For the Fourier method with the Hamming window (Figure 8b) and the Dolph–Chebychev method (Figure 8c), the effect of quantization appears for all levels of quantization. The design using the zero placement method (Figure 8d) (and particular positions of zeros) seems to be practically inapplicable.

Besides the quantization effect, we can see the influence of the method on the range of possible scanning angle in Figure 8. Note that the sensitivity to quantization of the excitation magnitudes and phases also points to sensitivity in general.

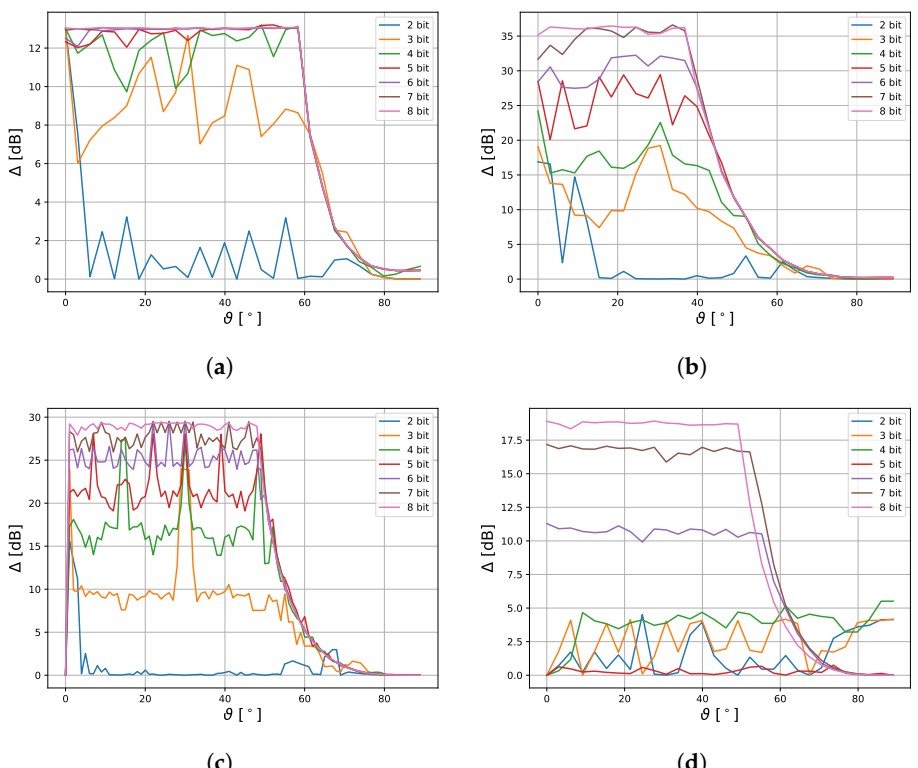

**Figure 8.** The influence of quantization of coefficients. (**a**) Fourier method with rectangular window; (**b**) Fourier method with Hamming window; (**c**) Dolph–Chebychev with required side lobe level −30 dB; and (**d**) Zero placement method.

Figure 9a,b show the influence of the required suppression level on zero positions. Depending on the specific application, a trade-off must be made between the required radiation pattern and sensitivity. For a rough quantization and high requirements on suppression of side lobes, the position of zeros is significantly influenced. On the other hand, the tuning of positions of zeros can be a way to decrease sensitivity to quantization.

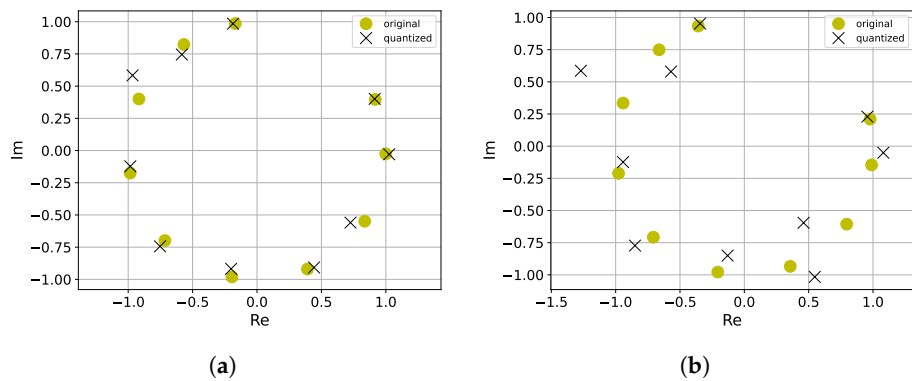

**Figure 9.** Influence of 3-bit quantization of coefficients on positions of zeros. (**a**) Dolph–Chebychev method 20 dB; (**b**) Dolph–Chebychev method 30 dB.

The synthesis of array factors based on the deterministic method certainly will not lead to an acceptable design. However, it can significantly reduce the number of evaluations of the goal function during the following optimization with the usage of field simulation.

For testing capabilities of different optimization algorithms, the problem of Array Factor synthesis where the goal function in the form

$$\mathcal{F}_1 = \max[S_a(\vartheta_r, \varphi_r)], \tag{24}$$

($\vartheta_r$ and $\varphi_r$ standing for the required elevation and azimuth, respectively) was chosen. As written above, the array factor obtained using this goal function is the same as the array factor obtained using the Fourier method with the rectangular window.

The optimization was performed for 200 individuals and 300 generations. The first generation was created purely randomly with the uniform distribution. After preliminary tests, the four evolutionary algorithms were chosen. The parameters of optimization were magnitudes and phases of all array elements independently. This can be useful during the calibration process, where symmetries can be broken due to possibilities of practical realization. The results of optimization are depicted in Figure 10. Figure 10a shows the dependence of array factor on elevation angle after optimization. The goal was to maximize the array factor on the elevation angle $\vartheta = 0$. Figure 10b shows the dependence of value of the array factor $S_a(0,0)$ on the number of generation. It can be seen that the fastest convergence was reached using the SMPSO algorithm over generations (*K*). However, the number of calculations of the goal function is too high for incorporating electromagnetic field simulation. The algorithm EPS MOEA does not converge to the correct solution at all.

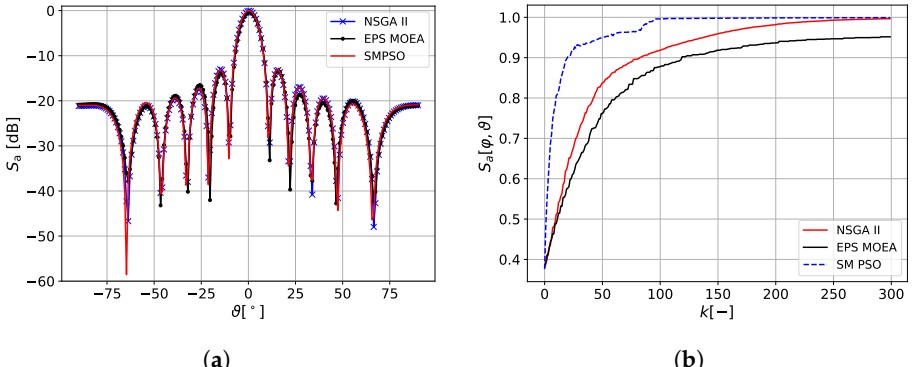

**Figure 10.** Comparison of algorithms NSGA II, EPS MOEA, and SMPSO. (**a**) Array factor; (**b**) Convergence.

Figure 11 shows the possibilities of speeding up the optimization process in the case of the algorithms NSGA II and SMPSO. Figure 11a shows the array factor obtained using optimization by different approaches, while Figure 11b shows the convergence of different sets of parameters. The fastest convergence seems to be reached if parameters of optimization are positions of zeros. Unfortunately, optimizing the zero positions often tends to converging to non-optimal solutions (see the red line in Figure 11b). The used objective function is not equally sensitive to the positions of all zeros. More robust convergence is observable when using polynomial coefficients as optimization parameters. The optimization using SMPSO with coefficients of polynomials as parameters is acceptably fast and seems to be applicable together with the field simulator. Although optimization of zero positions shows relatively poor convergence together with evolutionary algorithms, the change of zero position has a predictable impact on array factor (or radiation pattern). It makes the zero placing be a candidate for cooperating with machine learning for automatic tuning and calibration.

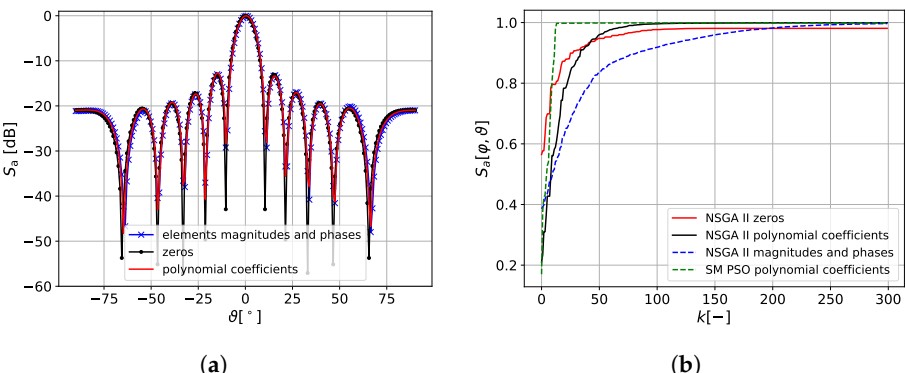

**Figure 11.** Convergence of algorithm with different parameter sets. (**a**) Array factor; (**b**) Convergence.

The number of enumerations of the objective function in the previous test problem would be on the edge of solvability if the field simulation is used. It is also necessary to mention that, for other objective functions, the convergence is mostly even worse. The most promising approach seems to be the set of parameters $\delta\alpha_1, \delta\alpha_2, \ldots, \delta\alpha_{n-1}$, where each parameter $\delta\alpha_i$ represents a relative angular shift of the particular zero. The optimization process therefore can start from the solution obtained using any deterministic method. In this work, the objective function was formulated as

$$\mathcal{F}_1 = \max |(S_{\mathrm{ar}i} - S_{\mathrm{a}i})|, \tag{25}$$

where $S_{\mathrm{ar}i}$ is the required magnitude of the $i$-th lobe and $S_{\mathrm{a}i}$ is the magnitude of the $i$-th lobe. The goal function described by Equation (25) leads to maximal suppression of the side lobes.

Any standard deterministic method does not take into account the sensitivity to quantization. Additionally, the single objective optimization presented above does not reduce the sensitivity.

One possible way to reach the prescribed radiation pattern and reduce the sensitivity to rounding (and sensitivity in general) at the same time, is represented by a multi-criteria optimization. First, the goal function $\mathcal{F}_1$ represents the requirements on the array factor shape, and is defined by equation (25). The second goal function represents the effect of quantization and is defined by

$$\mathcal{F}_1 = \max |(\hat{S}_{\mathrm{ar}i} - \hat{S}_{\mathrm{a}i})|, \tag{26}$$

where ($\hat{S}_{\mathrm{ar}i}$ and $\hat{S}_{\mathrm{a}i}$) are magnitudes of side lobes of array factor obtained using coefficients quantized by required number of bits.

Figure 12a shows all calculated solutions for multi-criteria optimization. A collection of solutions that are not dominated by other solutions (in the figure marked by red) in that set are superior to the rest of the solutions. In the search area, they are known as the Pareto front [37]. The red point at the rightmost position represents the solution in which requirement on array factor shape are best fulfilled; the red point at the leftmost position represents the solution which is less sensitive to quantization. The remaining red points represent the compromise between the required array factor shape and the sensitivity to rounding. The dependence of difference between the main lobe and the biggest side lobe for 3-bit quantization is depicted in Figure 12b. As can be seen from the Pareto front, it is possible to select a solution with an acceptable side lobe suppression and also acceptable sensitivity to quantization. Especially for quantization via a low number of bits, the proposed method offers better results than any tested deterministic method. Figure 12c,d show the comparison of the best deterministic methods and the proposed method for the array factor and required elevation angle $\vartheta = 0°$ and $\vartheta = 30°$, respectively.

Note that results were obtained using SMPSO algorithm which showed the best performance in the previous tests.

The important question is if the usage of multi-criteria optimization instead of mono-criteria optimization does not lead to significantly worse convergence. The comparison of convergence between the mono-criteria and multi-criteria processes is depicted in Figure 13a,b, respectively. The results were obtained using the SMPSO algorithm, where the whole optimization process was repeated 20 times. The figure shows the mean value and standard deviation calculated from these twenty runs for both single and multi-objective optimization. The usage of multi-criteria optimization has a certain but not critical influence on the convergence speed.

According to the results described above, the lowest number of calculations of the goal function was reached using procedure:

1.　Calculating solutions using deterministic methods to reach a result close to the required radiation pattern (array factor).
2.　Forming the initial population including results obtained using deterministic methods.
3.　Performing the multi-criteria optimization using the SMPSO algorithm, where parameters of optimization are relative shifts of the zero positions. The first goal function describes the required shape of the array factor, while the second goal function expresses the influence of the quantization error.
4.　Selecting an acceptable compromise between the array factor requirements and sensitivity to quantization from Pareto front obtained using optimization.

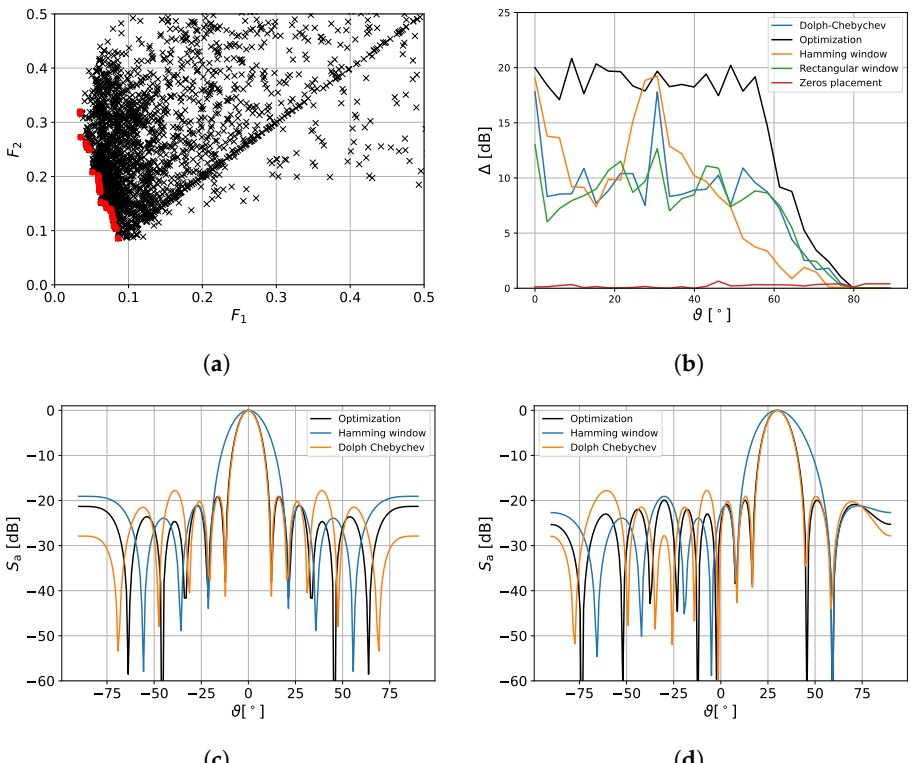

**Figure 12.** Results of multi-criteria optimization with 60 generations and 60 individuals in each generation. (**a**) Pareto front of the quantization error of the magnitude and the phase after the multi-criteria optimization, the points of the Pareto front is denoted by red dots; (**b**) Comparison of effect of 3-bit quantization for array factor obtained by different methods. The result of optimization—black line; (**c**) Comparison of array factors obtained by different methods with 3-bit quantized excitation for $\vartheta = 0°$; (**d**) Comparison of array factors obtained by different methods with 3-bit quantized excitation for $\vartheta = 30°$.

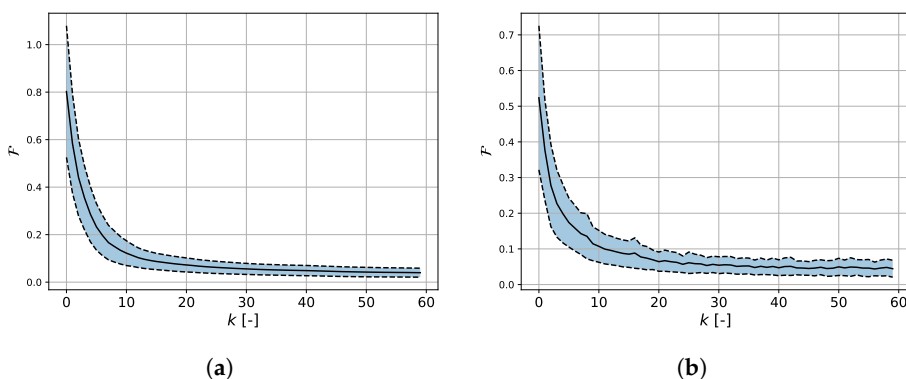

(**a**)                  (**b**)

**Figure 13.** Mean and standard deviation calculated from 20 runs of optimization. (**a**) Single-criteria optimization; (**b**) Multi-criteria optimization.

## 4. Conclusions

The main goal of this research, motivated by requirements following the necessity of communication with nanosatellites, was to find the procedure of synthesis of antenna array control. This procedure should take into account the sensitivity to quantization errors, but, at the same time, the number of necessary evaluations of the goal functions (which can be based on full-field simulation) must be minimized. It was shown that the commonly used deterministic methods could lead to control, which is highly sensitive to quantization, and the following optimization step proved necessary. On the other hand, including results from the deterministic approach to the initial populations of the optimization process rapidly reduces the number of calculations of the goal functions. The fastest convergence was reached using the algorithm SMPSO when the parameters of optimization were relative angular shifts of zeros positions.

**Author Contributions:** Conceptualization, D.P. and T.O.; methodology, D.P.; software, T.O., D.P., H.K.N. and P.K.; validation, D.C.D.G., H.K.N. and P.K.; writing—review and editing, D.P. and T.O. All authors have read and agreed to the published version of the manuscript.

**Funding:** This research was funded by Czech Science Foundation (GACR) grant number 20-02046S.

**Acknowledgments:** This work was supported by University of West Bohemia during an internal project SGS-2021-011.

**Conflicts of Interest:** The authors declare no conflict of interest.

## Abbreviations

The following abbreviations are used in this manuscript:

NSGA         Not-dominated sorting genetic algorithm
PSO           Partical Swarm Optimization
GA             Genetic algorithm
ACO          Anto Colony Optimization
SMPSO      Speed Constrained Partical Swarm Optimization
EPS-MOEA   Epsilon Multiobjective Evolutionary Algorithm
PCA          Principal Component Analysis
SVD          Singular Value Decomposition
SA            Simulated Annealing

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
