# Peer review of "Performance Comparison of Quantized Control Synthesis Methods of Antenna Arrays"

_electronics, doi:10.3390/electronics11070994_

Round 1

Reviewer 1 Report

This work seems to contain several interesting aspects. The analysis of the deterministic methods is clearly leading to the problem because of the quantization. Nevertheless, the major problem in this work is the optimization process analysis that is very poor since it lacks various important information. For example, the authors must mention explicitly the quantization level to obtain their results. There is a reference in Fig. 11b but it is not enough. Otherwise, the results seem identical to the deterministic solvers (Fig. 4 with Fig. 8a), and the novelty is in a question. Some additional important comments:

  1. The notation in most figures (and their explanation in the text) is very ambiguous. For example, it seems that in Figure 6, the parameter Delta corresponds to the level of sidelobes. However, there is not any reference to this.
  2. In general, the control of the feeding current's amplitude and phase can be continuous. Why is the quantization used?
  3. What is k in Figures 8b, 9b, and 10? It should be mentioned in the text.
  4. What is a Pareto front? At least a reference is required.
  5. The authors include in their work a conceptual diagram for the antenna design process. However, they should focus on a conceptual diagram that describes their methodology (as described briefly in the conclusion section).

Author Response

Dear Reviewer,

 The authors would like to thank the editor-in-chief and all the reviewers for the time they spent providing comments. We hope that our answers and the changes in the manuscript clarify all the questions and enhance the quality and value of the manuscript. An effort has been made to address all the concerns by the reviewers and to accommodate all their suggestions in order to enrich the revised manuscript.

Main Questions:

This work seems to contain several interesting aspects. The analysis of the deterministic methods is clearly leading to the problem because of the quantization. Nevertheless, the major problem in this work is the optimization process analysis that is very poor since it lacks various important information. For example, the authors must mention explicitly the quantization level to obtain their results. There is a reference in Fig. 11b but it is not enough. Otherwise, the results seem identical to the deterministic solvers (Fig. 4 with Fig. 8a), and the novelty is in a question.

Answer:

Thank you for your comments. The end of the Introduction section and the results and discussion section was rewritten to be more understandable for the readers, all of the changes highlighted by blue color in the manuscript. Section 2.6 was rewritten to better describe the applied techniques' level for considering the quantization. During the calculations, a 3-bit quantization level was considered. New figures and explanations were added to improve the understandability of the paper. Moreover, all of the data and the calculations are available for the interested researchers from the project's GitHub repository. 

The novelty and the main difference from the methods implemented in simulation programs do not take into account the influence of quantization error. The methods are based just on proper rounding of coefficients obtained by deterministic methods. Moreover, these methods can not include results of deterministic methods, numerical simulations to the initial population and they are don’t apply multicriteria optimization in order to ensure low sensitivity. The goal of this project is to design an antenna array for a satellite system, for this development process the paper contains a simple benchmark problem, similarly to Compumag’s TEAM benchmark problems. Due to the simplicity of the problem, it can be solved analytically and by numerical codes, as well. Therefore, the solution to this problem can ensure the correctness of more complex, numerical simulation-based calculations. All, of the data and the user codes, are available on the projects’ GitHub page.

  1. The notation in most figures (and their explanation in the text) is very ambiguous. For example, it seems that in Figure 6, the parameter Delta corresponds to the level of sidelobes. However, there is not any reference to this.

Answer:  The notations were improved.

2. In general, the control of the feeding current's amplitude and phase can be continuous. Why is the quantization used?

Answer:  The advantage of quantized control of antenna array is mainly in the possibility of fast and precise reconfiguration of the antenna array beam direction which is important for tracking satellite position. (lines 48 - 50)

3. What is k in Figures 8b, 9b, and 10? It should be mentioned in the text.

Answer: We accepted the comment. Now all figures are referenced in the text.

4. What is a Pareto front? At least a reference is required.

Answer: We accepted the comment. The meaning of this term is explained in lines 285 - 295 and the reference is added.

5. The authors include in their work a conceptual diagram for the antenna design process. However, they should focus on a conceptual diagram that describes their methodology (as described briefly in the conclusion section).

Answer: We accepted the comment.  The methodology was summarized at the end of the Discussion section.

Reviewer 2 Report

The paper discusses the problem of quantized control for antenna array. This task is relevant for connection with small satellites. Well-known analytical methods are summarized and stochastic optimization methods are considered.

The reviewer has two main remarks to the article:

  1. The goal of the comparison is to propose the methodology of antenna array synthesis using the opportunities of computer electromagnetics simulation. But in the article there isn’t results confirming the advantage of the proposed solution over existing ones, for example, those implemented in the simulation programs themselves. The additional simulation examples are needed.
  2. As a result of the work, a clear formulation of the optimization algorithm with the limitations and process parameters identified as a result of the comparison is not presented.

In addition, there are the following minor remarks:

  1. There is no diagram description in figure 1
  2. Relationship between (2) and (3) is not shown: (2) is scalar, but (3) is vector. In (3) it is not clear what are l and k. Indices m and n in (2) and (4) have been swapped.
  3. Formula (10) should be checked.
  4. «Three well-known algorithms are tested within the paper: genetic algorithms NSGAII, EPS-MOEA and swarm optimisation algorithm SMPSO». Why were these particular algorithms chosen?
  5. Formulas (18) and (19) require explanation
  6. In figure 6 (d) the line for 3 bits looks very strange (approximately equal for all angles)

Author Response

Dear Reviewer,

 The authors would like to thank the editor-in-chief and all the reviewers for the time they spent providing comments. We hope that our answers and the changes in the manuscript clarify all the questions and enhance the quality and value of the manuscript. An effort has been made to address all the concerns by the reviewers and to accommodate all their suggestions in order to enrich the revised manuscript.

  1. The goal of the comparison is to propose the methodology of antenna array synthesis using the opportunities of computer electromagnetics simulation. But in the article there isn’t results confirming the advantage of the proposed solution over existing ones, for example, those implemented in the simulation programs themselves. Additional simulation examples are needed.

Answer: Thank you for your comments. The Introduction, Conclusion section and the results and discussion section was rewritten to show the more clear contribution and to be more understandable for the readers, all of the changes highlighted by blue color in the manuscript.

The novelty and the main difference from the methods implemented in simulation programs do not take into account the influence of quantization error. The methods are based just on proper rounding of coefficients obtained by deterministic methods. Moreover, these methods can not include results of deterministic methods, numerical simulations to the initial population and they are don’t apply multicriterial optimization in order to ensure low sensitivity. The goal of this project is to design an antenna array for a satellite system, for this development process the paper contains a simple benchmark problem, similarly like Compumag’s TEAM benchmark problems. Due to the simplicity of the problem, it can be solved by analytically and by numerical codes, as well. Therefore, the solution of this problem can ensure the correctness of more complex, numerical simulation based calculations. All, of the data and the used codes available in the projects’ github page.

2. As a result of the work, a clear formulation of the optimization algorithm with the limitations and process parameters identified as a result of the comparison is not presented.

Answer: We accepted the comment and tried to improve the Discussion and Conclusion sections.

In addition, there are the following minor remarks:

3. There is no diagram description in figure 1

Answer: We accepted the comment. According to the comment from another reviewer the Figure 1 was removed.

4. Relationship between (2) and (3) is not shown: (2) is scalar, but (3) is vector. In (3) it is not clear what l and k. Indices m and n in (2) and (4) have been swapped.

Answer: We accepted the comment. Mentioned part of the text was rewritten (lines 131 - 137) .

5. Formula (10) should be checked.

Answer: Formula checked.

6. «Three well-known algorithms are tested within the paper: genetic algorithms NSGA-II, EPS-MOEA and swarm optimization algorithm SMPSO». Why were these particular algorithms chosen?

Answer: We accepted the comment. The reasons are commented in lines: 196 - 203. 

7. Formulas (18) and (19) require explanation

Answer: We accepted the comment. We tried to explain the meaning of these equations in paragraph 2.6 (lines 206-2011).

8. In figure 6 (d) the line for 3 bits looks very strange (approximately equal for all angles)

Answer: We agree with this comment. This is true for all quantizations worse than 4 bits. It means that 3-bit quantization can not be used for these particular zeros positions.

Reviewer 3 Report

Point 1: Make more clear the contribution of the authors in terms of novelty of this paper. There are comparisons between quantized control synthesis
methods of antenna arrays, but the conclusions don't discuss too much the results.

Point 2: For the example, a 11 × 11 antenna array was chosen for analyze. Can the authors make a comment on how the dimension of the array was chosen? How does a larger or a smaller array affect the results?

Point 3: A more detailed interpretation of the results in Section 3 should be useful. Also, measurement results should be used to compare the simulations with real-life scenarios. 

Point 4: Conclusion section must be improved on the above comments. 

Author Response

Dear Reviewer,

The authors would like to thank the editor-in-chief and all the reviewers for the time they spent providing comments. We hope that our answers and the changes in the manuscript clarify all the questions and enhance the quality and value of the manuscript. An effort has been made to address all the concerns by the reviewers and to accommodate all their suggestions in order to enrich the revised manuscript.

Point 1: Make more clear the contribution of the authors in terms of novelty of this paper. There are comparisons between quantized control synthesis methods of antenna arrays, but the conclusions don't discuss the results too much.

Answer: Thank you for your comments. The Introduction, Conclusion section and the results and discussion section was rewritten to show the more clear contribution and to be more understandable for the readers, all of the changes highlighted by blue color in the manuscript.

 The novelty and the main difference from the methods implemented in simulation programs do not take into account the influence of quantization error. The methods are based just on proper rounding of coefficients obtained by deterministic methods. Moreover, these methods can not include results of deterministic methods, numerical simulations to the initial population and they are don’t apply multicriterial optimization in order to ensure low sensitivity. The goal of this project is to design an antenna array for a satellite system, for this development process the paper contains a simple benchmark problem, similarly like Compumag’s TEAM benchmark problems. Due to the simplicity of the problem, it can be solved by analytically and by numerical codes, as well. Therefore, the solution of this problem can ensure the correctness of more complex, numerical simulation based calculations. All, of the data and the used codes available in the projects’ github page.

Point 2: For the example, a 11 × 11 antenna array was chosen for analysis. Can the authors make a comment on how the dimension of the array was chosen? How does a larger or a smaller array affect the results?

Answer: The size of array is in accordance with the aim to replace commonly used parabolic antenna ie. to have a comparable gain and a comparable beam width. We commented on lines 113 - 126.

Point 3: A more detailed interpretation of the results in Section 3 should be useful. Also, measurement results should be used to compare the simulations with real-life scenarios. 

Answer: We accepted the comment and rewrote the Discussion section. Unfortunately the experiment is not possible to perform at the moment.

Point 4: Conclusion section must be improved on the above comments. 

Answer: We accepted the comment and rewrote the Conclusion section.

Round 2

Reviewer 1 Report

The scope of the paper is now clearer (the quantization is taken into account in the proposed method in contrast to the rounding of coefficients) but there are many details missing and the comparison is not fair at all.

  1. Let's consider in Fig.8 that the y-axis corresponds to the side-lobe level (there is still not any explanation for Delta [dB] of the y-axis in the text). The level is over 20dB until 40 degrees (for Fig.8b) and over 15db until 55 degrees for 3-bit quantization. The proposed result (Fig. 12b) is approximately 18dB for the same 3-bit quantization level for 0 degrees. Consequently, where is the optimization?
  2. The best side-lobe suppression level in this paper is presented in Fig.7b (over 40dB). The authors should compare their methodology with this result.
  3. The degradation is severe in Fig. 8 as the angle increases but the angle is not mentioned in the proposed method. Consequently, there is not any clue that the proposed methodology fix angle degradation.
  4. The optimization must be directly compared to the deterministic solvers. For example, 3-bit quantization at different angles for the Fourier method with Hamming window and the proposed method.
  5. The 3-bit quantization during calculations is still not mentioned in the text explicitly.

Author Response

Dear Reviewer, 

Thank you very much for your valuable comments, which allow us to improve our work. 

Let's consider in Fig.8 that the y-axis corresponds to the side-lobe level (there is still not any explanation for Delta [dB] of the y-axis in the text). The level is over 20dB until 40 degrees (for Fig.8b) and over 15db until 55 degrees for 3-bit quantization. The proposed result (Fig. 12b) is approximately 18dB for the same 3-bit quantization level for 0 degrees. Consequently, where is the optimization?

Answer:

We have to apologize for the mistake. The problem was with the scale in Fig. 8. The scale started from “0 bits”, which is obviously nonsense. After correcting the scale and regenerating all graphs in Fig. 7 and Fig. 8, we believe that results make more sense and the advantage of optimization is noticeable. The symbol Delta is now explained in lines 224 - 230.

Thank you very much for making us resolve this serious problem.

The best side-lobe suppression level in this paper is presented in Fig.7b (over 40dB). The authors should compare their methodology with this result.

Answer:

Our point was that the optimized solution seems better for a lower number of bits. Decreasing the number of bits reduces the number of phase shifters, which can be advantageous for larger arrays.

The degradation is severe in Fig. 8 as the angle increases, but the angle is not mentioned in the proposed method. Consequently, there is not any clue that the proposed methodology fix angle degradation.

Answer:

We added figures 12c and 12d, which show the relatively good stability of our solution along the elevation angle (comparing to methods in Figure 8).

The optimization must be directly compared to the deterministic solvers. For example, 3-bit quantization at different angles for the Fourier method with Hamming window and the proposed method.

Answer:

Now we believe that comparison of Fig. 7, 8 with Fig. 12. demonstrates the advantage of the proposed method.

The 3-bit quantization during calculations is still not mentioned in the text explicitly.

Answer: The 3-bit quantization is now mentioned in lines 299 - 303.

Reviewer 2 Report

I agree with most of the authors' corrections and consider that the article can be published.

Author Response

Dear Reviewer,

 Thank you for your positive feedback.

Reviewer 3 Report

The authors considered the comments and revised accordingly the manuscript.

Author Response

(The authors gave the same response as above.)

Round 3

Reviewer 1 Report

Most changes have been conducted, but the direct comparison is still avoided.

  1. Figure 8 in the previous version started from 0-bit that makes no sense, indeed. However, the new curves start from 2-bit, while there is one curve less. Moreover, the results for the Fourier method with Hamming window seem worst now (eg 'n bits = 2'). Consequently, it is not obvious that this Figure is now correct.
  2. The new Figures 12c and 12d are showing the behavior until 30 degrees. However, the deterministic methods suggest that the degradation occurs after 40 degrees. Therefore, the investigation must be performed for larger angles, too.
  3. The methods are compared via curves of different Figures; thus, it is difficult to detect the optimization. A direct comparison is required.

Concluding, the authors must include two Figures in order to highlight their method's enhancement. Specifically: one figure theta vs Delta with at least two curves (the proposed method and the best of deterministic using the same level of quantization). Additionally, Fig 12b must include the result of the best deterministic method with the same quantization level.

If such results are not included the scientific soundness will remain unacceptable since the optimized features of the proposed methodology will remain in question.

Author Response

Dear Reviewer,

Thank you very much for your valuable comments, which allow us to improve our work.

Most changes have been conducted, but the direct comparison is still avoided. 

1. Figure 8 in the previous version started from 0-bit that makes no sense, indeed. However, the new curves start from 2-bit, while there is one curve less. Moreover, the results for the Fourier method with Hamming window seem worst now (eg 'n bits = 2'). Consequently, it is not obvious that this Figure is now correct. 

The number of bits was changed to be the same in Fig.  7 and Fig. 8.  One bit quantization is unusable for all methods therefore it is omitted. 

2. The new Figures 12c and 12d are showing the behavior until 30 degrees. However, the deterministic methods suggest that the degradation occurs after 40 degrees. Therefore, the investigation must be performed for larger angles, too. 

The goal of the paper is to propose a robust and reliable methodology to design an antenna array for nanosatellites, not a general antenna design methodology. Here, the requirement is to examine the maximum angle till 30° (section 2.1), after this point the satellite can not observable, and the comparison is useless from the  application point of view.  

3. The methods are compared via curves of different Figures; thus, it is difficult to detect the optimization. A direct comparison is required. 

Concluding, the authors must include two Figures in order to highlight their method's enhancement. Specifically: one figure theta vs Delta with at least two curves (the proposed method and the best of deterministic using the same level of quantization). Additionally, Fig 12b must include the result of the best deterministic method with the same quantization level. 

Figures 12b - 12d were changed to allow direct comparison as suggested.